# The efficacy of interventions in reducing belief in conspiracy theories: A systematic review

Cian O'Mahony[1]*, Maryanne Brassil[2], Gillian Murphy[1], Conor Linehan[1]

1 School of Applied Psychology, University College Cork, Cork, Ireland, 2 School of Psychology, University College Dublin, Dublin, Ireland

* cianomahony@ucc.ie

## Abstract

Conspiracy beliefs have become a topic of increasing interest among behavioural researchers. While holding conspiracy beliefs has been associated with several detrimental social, personal, and health consequences, little research has been dedicated to systematically reviewing the methods that could reduce conspiracy beliefs. We conducted a systematic review to identify and assess interventions that have sought to counter conspiracy beliefs. Out of 25 studies (total N = 7179), we found that while the majority of interventions were ineffective in terms of changing conspiracy beliefs, several interventions were particularly effective. Interventions that fostered an analytical mindset or taught critical thinking skills were found to be the most effective in terms of changing conspiracy beliefs. Our findings are important as we develop future research to combat conspiracy beliefs.

## Introduction

Conspiracy beliefs are defined as a set of beliefs that "explain important events as secret plots by powerful and malevolent groups" [1], p. 538]. For example, various groups of individuals hold the beliefs that the 1969 moon landing was a hoax staged by the United States government, and that the cure to cancer has been discovered but is kept secret from the public [2,3]. Holding conspiracy beliefs is often associated with negative personal, social, and health-related consequences. For example, conspiracy beliefs are associated with reluctance to receive a COVID-19 vaccine and reduced adherence to public health regulations [4–9], extremist and violent behavior [10,11]. As much of this research shows only associations between conspiracy beliefs and negative outcome, further research is needed to draw any conclusions regarding causality. Nevertheless, as there are possible negative implications associated with conspiracy beliefs at both personal and societal levels, there is a need for interventions to lower the likelihood that people will engage uncritically with such theories.

One of the primary characteristics of a conspiracy theory is that they are unfalsifiable. Specifically, any attempt to demonstrate that the claims made by conspiracy theories are false, could be perceived as evidence of a cover-up of the "truth", and institutions that attempt to debunk conspiracies are perceived as accomplices attempting to conceal such conspiracies [12,13]. The effectiveness of counterarguments on conspiracy beliefs is mixed. Several studies

**Data Availability Statement:** The data collected and analysed in this study can be found on the Open-Science Framework (https://osf.io/6yjt3/).

**Funding:** Awardee: COM Grant Number: EPSPG/2021/212 Funder: Irish Research Council URL:

https://research.ie/ The funders had no role in study design, data collection and analysis, decision to publish, or preparation of the manuscript.

**Competing interests:** Google, in association with the Irish Research Council, funded this study, but this does not alter our adherence to PLOS ONE policies on sharing data and materials. There are no patents, products in development or marketed products associated with this research to declare.

have found that counterarguments could change the conspiracy beliefs of participants [14]. However, other studies found that counterarguments had no effect of conspiracy attitudes [15,16]. Additionally, some research has found that counterarguments in some particular circumstances can paradoxically strengthen conspiracy beliefs [17–19]. As such, simply arguing against conspiracy beliefs may not be a sufficient way of changing them.

## Conspiracy interventions

The following section introduces and defines the various types of interventions that have been designed to influence conspiracy belief. It should be noted that much of the following research surrounding conspiracy interventions were designed based on evidence accumulated by studies that focus on misinformation and fake news in addition to conspiracy belief. Such papers frequently refer to misinformation, fake news, and conspiracy beliefs interchangeably when citing evidence to support the utility of a particular intervention. While the authors of this paper believe that misinformation, fake news, and conspiracy beliefs are distinct phenomena, the common characteristic of the studies in this area is that they attempt to change beliefs that persist, even when sufficient counterevidence exists.

The first category of conspiracy interventions are *informational inoculations*. The theory behind this intervention suggests that much like a vaccine protects against disease, an informational inoculation consisting of a pre-emptive debunking [20]. The inoculation consists of giving participants a weakened form of the conspiracy argument along with points that refute the claims made in the argument. So, whereas a direct counterargument or debunking attempts to confront conspiracy beliefs after people have been exposed to conspiracy theories, an informational inoculation confronts conspiracy beliefs before exposure. Previous experiments have shown these inoculations are effective with regards to decreasing belief in fake news and political gossip, as well as conspiracy beliefs [20–23]. A meta-analysis has also shown that inoculations overall represent an effective means of challenging belief persistence, in situations where participants would otherwise be unwilling to change their beliefs despite contradictory evidence [24].

Another common type of intervention for changing conspiracy beliefs is *priming*. Priming interventions experimentally manipulate the psychological states of participants before they are asked about their conspiracy beliefs. For example, Swami et al [20b], found that priming participants to engage in analytical thinking significantly changed the likelihood that they would agree with conspiracy beliefs when tested. While priming interventions have shared features (i.e. they expose participants to a task or stimulus with the intention of subsequently influencing how they score on conspiracy measures), they are a heterogenous group. Some priming interventions seek to alter one's cognition by administering tasks that increase focus and analytical thinking [25]. Other priming interventions sought to influence one's affective state, for example altering their feelings of ostracism [26]. Some researchers consider priming to be another form of informational inoculation, as they also consist of treatments that are administered before participants are exposed to conspiracy theories [27].

Other approaches attempt to use the persuasive power of stories to confront conspiracy beliefs, often referred to as *narrative persuasion* [28]. The hypothesis is that the narrative and anecdotal nature of conspiracy theories are more appealing than the detached nature of scientific evidence. These claims have been supported by the fact that personal anecdotes both in favour and against vaccines were found to be far more persuasive than scientific evidence [29].

Conspiracy beliefs have been associated with a number of cognitive biases and logical fallacies such as the conjunction fallacy [30], jumping-to-conclusions bias [31], and illusionary pattern perception [32]. These biases have been attributed to the unfounded conclusions drawn by those who hold conspiracy beliefs. A number of the interventions similar to those listed

above, such as fostering effortful thought, have been successful in reducing susceptibility to some of the cognitive biases associated with conspiracy belief (e.g. conjunction fallacy) [33]. However, it is difficult to draw conclusions regarding how these interventions change the underlying cognitive processes behind conspiracy belief, as many of these studies exclusively measure the extent in which people endorse conspiracy theories.

### Current study

Systematic reviews within the area of conspiracy research have typically focused on identifying and understanding the individual differences and predictors of conspiracy belief [1,34]. There is yet to be a comprehensive review of the efficacy of various conspiracy interventions. While one systematic review focused on examining conspiracy interventions, it was solely focused on the use of narrative persuasion in the specific case of anti-vaccine conspiracy theories [28]. As there are growing calls to effectively combat widespread conspiracy beliefs, it is pertinent that we review the evidence for existing interventions. The COVID-19 pandemic and associated conspiracy theories highlighted the potentially harmful public health threats posed by unsubstantiated beliefs. By reviewing the evidence, we may gain a greater understanding of how to challenge conspiracy beliefs.

## Methods

The protocol was written in accordance to the PRISMA-P 2015 Statement guidelines (Moher et al., 2015). PRISMA is a 27-point checklist of the preferred reporting items for a systematic review (see S1 File). The authors pre-registered this review at PROSPERO (https://www.crd.york.ac.uk/prospero/display_record.php?ID=CRD42021256263).

### Search strategy

Three electronic databases were searched (PubMed, PsychIINFO, Scopus). The search strategy of this review used two search categories; conspiracy theories, and reduction/intervention. The search strings were any combination of (conspirac* OR 'unsubstantiated beliefs' OR 'implausible beliefs' OR 'unsubstantiated claims' OR 'unfounded beliefs' OR truther) AND (intervention OR reduc* OR chang* OR alter* OR experiment OR persua*). The list of keywords was determined after repeated prechecking.

   With the concern of missing potential eligible papers, no filtering was applied in the database searches (subject, publication dates) to avoid the possibility of the scope of the view becoming too narrow. There was no limitation applied to the publication dates of papers in the electronic search. Searches were conducted on 3rd June 2021.

### Eligibility criteria

The PICO model [35] was used to determine the eligibility criteria for the current review. During a preliminary review of the literature, we found that very few papers had been published on the topic of conspiracy interventions. Furthermore, literature concerning conspiracy beliefs was found to use various terms to refer to such beliefs (e.g., 'empirically unfounded beliefs', 'conspiratorial ideation' etc.). Lastly, the research designs of studies researching conspiracy interventions vary considerably. As such, the authors concluded that using a strict inclusion/exclusion criteria would exclude potentially useful papers from this review, and opted to use a broad inclusion criteria to mitigate this possibility.

   The inclusion criteria for this review consisted of the following:
   ***Population:***

- Adults above the age of 18

- Participants without clinical/psychiatric afflictions

    *Intervention:*

- Includes a measure of conspiracy belief (both general and specific measures were accepted).

- Experimental condition/stimulus used with the intention of influencing the measurement of conspiracy belief.

    *Comparator:*

- A control condition either between or within-groups

    *Outcomes:*

- Difference in conspiracy belief between experimental and control conditions

    The exclusion criteria for this review consisted of the following:
    *Population:*

- Participants under 18 years of age

- Clinical populations

    *Intervention:*

- Correlation studies/observational measures.

    *Comparator:*

- Studies with no control/baseline condition for the experimental intervention condition

    *Outcomes:*

- No outcomes were excluded from this study.

## Data management

References were collected from PsychInfo, PubMed, and Scopus. References were then imported into Endnote, where they were screened for possible duplicates. Once the duplicates were removed, the remaining references were exported to Covidence.

## Data review and screening

The screening process was conducted using the Covidence systematic review management online software. The screening process consisted of two waves., Title and Abstract screening, and Full-text screening, in accordance to PRISMA standards [36]. Two reviewers (COM & MB) screened the papers for both the Title/Abstract, and Full Text data screenings. Conflicting votes were resolved by a third reviewer (GM) who made the final decision regarding the inclusion of such papers. The interrater reliability was assessed using Cohen's Kappa (k). The level of agreement was found to be fair (k = 0.40) for title and abstract screening, and substantial for full-text screening (k = 0.62). Initially, 16 papers passed the full-text screening. However, due to the broad nature of the inclusion criteria, a number of studies were deemed eligible but did not have sufficient data for analysis. As such, a second examination of the remaining papers was conducted to see whether each paper had sufficient data to analyse. Studies that reported neither sufficient data to calculate the Cohen's *d*, nor statistical significance, were excluded

from the analysis. One paper [37] was excluded as the paper reported its numerical findings solely in the form of graphs, making it difficult to extract exact values. Two papers were excluded as no relevant information could be obtained for our analysis. After the papers were screened for sufficient data, 13 articles in total were included in the final analysis. The PRISMA diagram (Fig 1) shows each stage of this process.

## Data extraction and analysis

A number of papers included in the review consisted of more than a single study, leading to a total of 24 studies being included in the final review. The following data was extracted from the studies: (1) Publication characteristics such as author names, journal, and publication dates; (2) Demographic characteristics, such as mean age, country, target population; and (3) Intervention characteristics–type of intervention used, conspiracy measure used, whether the conspiracy measure was specific of generic.

From the 25 studies included in our review, we extracted 48 intervention comparisons ($k$) in total (see Table 1). The outcome measure of interest was the Cohen's $d$ effect size. For all papers that used paired group designs, the Cohen's $d$ had already been reported. Where the Cohen's $d$ was not reported in a study, we calculated the $d$ value for the independent group designs using the mean difference and the pooled standard deviation [38]. The Cohen's $d$ was calculated for each comparison using Microsoft Excel. Where there was insufficient data to calculate the $d$ value, statistical significance was used to indicate whether the intervention successfully induced any change in belief. Quality assessment was conducted using the Risk of Bias Tool 2 (Sterne et al., 2019, see S1 Fig) [39]. To avoid repetition, the Risk of Bias was applied to the 25 studies as opposed to each comparator, as each comparator was subject to same randomisation process and measurement within their respective studies.

## Results

### Characteristics of studies

Out of the 13 articles included, the majority were conducted in English speaking countries; US (3), UK (3). The other studies were conducted in France (2), Hungary (1), Macedonia (1), Hong Kong (1), and Belgium (1). It should be noted that since the majority of studies were conducted online, a number of the studies did not use samples from their countries of origin. Out of the three studies conducted in the UK, two consisted of samples from the United States. Furthermore, the lone study conducted in Hong Kong also used participants from the United States. All studies were published between 2013 and 2020. As highlighted by the Risk of Bias assessment (see S1 Fig), most studies were not preregistered. All papers were peer-reviewed, and had English language versions available.

This review aimed to identify the various interventions that have been studied as a means of challenging conspiracy beliefs. We identified a number of different intervention types within the review. The first type of interventions were informational inoculations. These inoculation interventions had two different variations; one drew attention to the factual inaccuracies of conspiracy theories, and one that described logical fallacies of thought process underpinning conspiracy beliefs [15,21]. The second type of intervention identified was priming. As with inoculation interventions, priming interventions also had several variations. One variation used priming as means of manipulating participants sense of control [44]. These control priming interventions were typically coupled with a second priming task to examine how feelings of control interacted with other variables. Another variation of priming interventions consisted of inducing an analytical mindset in participants by having them complete certain tasks that required mental acuity (e.g., reading difficult-to-read fonts) [25]. The third type of

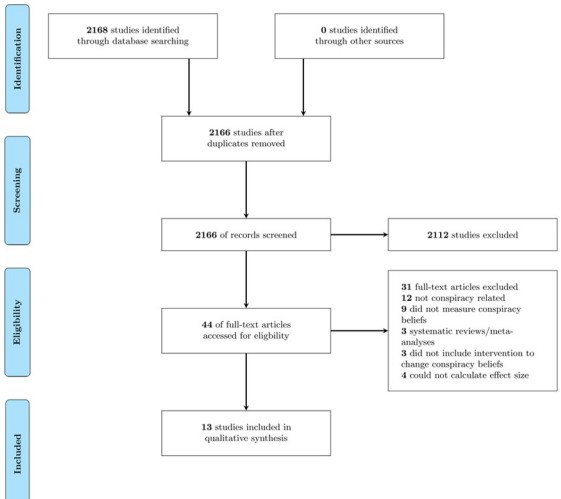

**Fig 1. PRISMA flow diagram.**

intervention identified were counterargument interventions. These interventions took the approach of providing counterarguments to conspiracy beliefs. A number of variations included arguments that appealed to the participants sense of empathy by describing the negative consequences conspiracy beliefs can have [42]. Another attempted to argue against conspiracy beliefs by ridiculing those who believe in them [42]. Finally, a number of standalone interventions were found in our sample, which did not fit into any of the intervention types described above, such as an educational intervention and a labelling intervention. An overview of the characteristics of each study in the review is provided in S1 Table.

## Efficacy of interventions

**Overview of included interventions..** A complete overview of the 48 comparisons analysed within this review, along with their the effect sizes and *p*-values is provided in Table 1. All of the comparisons analysed in this review used self-reported measures of conspiracy belief. Conspiracy belief measures fall into one of two categories; general measures and specific measures. General measures measure an individual's general disposition to conspiracy ideation based on the extent in which they endorse conspiratorial themes (e.g., that certain events are the result of small malicious group who control the world) [47]. Specific measures assess the extent in which people endorse specific conspiracy theories, such as conspiracy theory suggesting that the September 11 attacks were orchestrated by the US government [48]. As these two types of measures vary, it should be noted that the implications of decreasing scores on a specific measure compared to a general measure may differ [49,50]. Furthermore, the correlates for both types of measures differ. An intervention that changes how one endorses anti-vaccination conspiracy theories may not necessarily affect how they score on a measure of general conspiracy ideation. As noted previously [49], general measures of conspiracy beliefs are more stable and less malleable than their specific measure counterparts.

Approximately 60% of studies used some form of measure that examined generalised conspiracy belief and ideation ($k = 28$), while the rest used measures that assessed belief in specific conspiracy theories ($k = 6$). Only a small number of studies utilised a within-group design ($k = 5$) meaning the majority of comparisons used cross-sectional designs. Overall, 21 (~40%) interventions successfully changed conspiracy beliefs when compared to control, baseline, or

**Table 1. Effectiveness of interventions in included studies.**

| Author | Intervention | Design | Grouping | Aware of Intervertion | Conspiracy measure | Type of Measure | *p-value* | *Cohen's d* |
|---|---|---|---|---|---|---|---|---|
| [40] | Prevention Regulatory Focus | Experimental | Between-groups | Unclear | Cooperative Congressional Election Study (Ansolabeher, 2013) | General | 0.132 | 0.132 |
| [40] | Promotion Regulatory Focus | Experimental | Between-groups | Unclear | Cooperative Congressional Election Study (Ansolabeher, 2013) | General | 0.014* | 0.37 |
| [40] | Increasing Perceived Control (Promotion) | Experimental | Between-groups | Unclear | Cooperative Congressional Election Study (Ansolabeher, 2013) | Specific | 0.017* | 0.49 |
| [40] | Increasing Perceived Control (Prevention) | Experimental | Between-groups | Unclear | Cooperative Congressional Election Study (Ansolabeher, 2013) | Specific | 0.534 | 0.11 |
| [41] | Rationality Priming | Experimental | Between-groups | Yes | Conspiracist Mentality Questionnaire (Lantian, Muller, Nurra, & Douglas, 2016). | General | NR | -0.016 |
| [42] | Ridiculing Beliefs | Experimental | Within-groups | Yes | Conspiracy Assessment Tool (CAT) | General | 0.001** | 0.11 |
| [42] | Rational counterarguments | Experimental | Within-groups | Yes | Conspiracy Assessment Tool (CAT) | General | 0.001** | 0.13 |
| [42] | Empathetic counterarguments | Experimental | Within-groups | Yes | Conspiracy Assessment Tool (CAT) | General | 0.09 | 0.05 |
| [42] | Ridiculing Beliefs | Experimental | Between-groups | Yes | Conspiracy Assessment Tool (CAT) | General | 0.059 | 0.2 |
| [42] | Rational counterarguments | Experimental | Between-groups | Yes | Conspiracy Assessment Tool (CAT) | General | 0.011* | 0.27 |
| [42] | Empathetic counterarguments | Experimental | Between-groups | Yes | Conspiracy Assessment Tool (CAT) | General | 0.333 | 0.1 |
| [21] | Fact-based Inoculation | Experimental | Between-groups | No | Generic attitudes (Burgoon, Cohen, Miller, & Montgomery, 1978) | Specific | <0.001 | 1.311 |
| [21] | Logic-based Inoculation | Experimental | Between-groups | No | Generic attitudes (Burgoon, Cohen, Miller, & Montgomery, 1978) | Specific | <0.001 | 0.909 |
| [21] | Metainoculation (fact based)[2] | Experimental | Between-groups | No | Generic attitudes (Burgoon, Cohen, Miller, & Montgomery, 1978) | Specific | NR | 0.58 |
| [21] | Metainoculation (logic based)[2] | Experimental | Between-groups | No | Generic attitudes (Burgoon, Cohen, Miller, & Montgomery, 1978) | Specific | NR | 0.58 |
| [25] | Analytical priming | Experimental | Between-groups | Unclear | Belief in Conspiracy Theories Inventory (BCTI Swami et al., 2010, 2011). | General | 0.018* | 0.46 |
| [25] | Analytical priming | Experimental | Between-groups | Unclear | Belief in Conspiracy Theories Inventory (BCTI Swami et al., 2010, 2011). | General | 0.001** | 0.49 |
| [25] | Analytical priming (Font) | Experimental | Between-groups | Unclear | 7/7 bombings questionnaire | Specific | 0.046* | 0.34 |
| [25] | Analytical priming (Font) | Experimental | Between-groups | No | Generic Conspiracist Beliefs scale (GCBS; Brotherton, French, & Pickering, 2013) | General | 0.014* | 0.42 |
| [15] | Pro-conspiracy arguments[1] | Quasi-Experimental | Between-groups | Yes | Anti-vaccine Conspiracy Beliefs Measure | Specific | 0.001** | -0.66 |
| [15] | Anti-conspiracy arguments | Quasi-Experimental | Between-groups | Yes | Anti-vaccine Conspiracy Beliefs Measure | Specific | 0.017* | 0.42 |
| [15] | Anti-conspiracy/ conspiracy arguments | Quasi-Experimental | Between-groups | Yes | Anti-vaccine Conspiracy Beliefs Measure | Specific | 0.047* | -0.1 |
| [15] | Conspiracy/anti-conspiracy arguments[1] | Quasi-Experimental | Between-groups | Yes | Anti-vaccine Conspiracy Beliefs Measure | Specific | 0.263 | -0.39 |
| [15] | Anti-conspiracy/ conspiracy arguments | Quasi-Experimental | Between-groups | Yes | Anti-vaccine Conspiracy Beliefs Measure | Specific | 0.1 | 0.62 |
| [15] | Conspiracy/anti-conspiracy arguments[1] | Quasi-Experimental | Between-groups | Yes | Anti-vaccine Conspiracy Beliefs Measure | Specific | 0.001** | 0.31 |
| [43] | Conspiracy labelling | Experimental | Between-groups | No | Generic Conspiracist Beliefs scale (GCBS; Brotherton, French, & Pickering, 2013) | General | >0.05 | NR |

*(Continued)*

**Table 1.** (Continued)

| Author | Intervention | Design | Grouping | Aware of Intervertion | Conspiracy measure | Type of Measure | p-value | Cohen's d |
|--------|-------------|--------|----------|----------------------|-------------------|----------------|---------|-----------|
| [43] | Conspiracy labelling | Experimental | Between-groups | No | Historical items | General | >0.05 | -0.1 |
| [43] | Conspiracy labelling | Experimental | Between-groups | No | Adaptation of Wood et al, 2012 | Specific | >0.05 | 0.07 |
| [27] | Priming Resistance to Persuasion | Experimental | Between-groups | No | Generic Conspiracist Beliefs scale (GCBS; Brotherton, French, & Pickering, 2013) | General | 0.012* | 0.56 |
| [27] | Priming Resistance to Persuasion | Experimental | Between-groups | No | Generic Conspiracist Beliefs scale (GCBS; Brotherton, French, & Pickering, 2013) | General | 0.029* | 0.3 |
| [27] | Priming Resistance to Persuasion | Experimental | Between-groups | No | Generic Conspiracist Beliefs scale (GCBS; Brotherton, French, & Pickering, 2013) | General | 0.004* | 0.37 |
| [16] | Debunking | Experimental | Between-groups | Yes | Generic Conspiracist Beliefs scale (GCBS; Brotherton, French, & Pickering, 2013) | General | 0.07 | 0.29 |
| [16] | Debunking, motives, fallacy | Experimental | Between-groups | Yes | Generic Conspiracist Beliefs scale (GCBS; Brotherton, French, & Pickering, 2013) | General | <0.05* | 0.37 |
| [44] | High control priming | Experimental | Between-groups | No | Conspiracy Theory Ideation subscale of Conspiracy Mentality Scale (Stojanov & Halberstadt) | General | NR | 0.11 |
| [44] | Low control priming[1] | Experimental | Between-groups | No | Conspiracy Theory Ideation subscale of Conspiracy Mentality Scale (Stojanov & Halberstadt) | General | 0.07 | 0.27 |
| [44] | High control priming | Experimental | Between-groups | Unclear | Conspiracy Theory Ideation subscale of Conspiracy Mentality Scale (Stojanov & Halberstadt) | General | >0.05 | -0.13 |
| [44] | Low control priming[1] | Experimental | Between-groups | Unclear | Conspiracy Theory Ideation subscale of Conspiracy Mentality Scale (Stojanov & Halberstadt) | General | >0.05 | -0.15 |
| [44] | Low control priming[1] | Experimental | Between-groups | No | Conspiracy Theory Ideation subscale of Conspiracy Mentality Scale (Stojanov & Halberstadt) | General | 0.35 | -0.13 |
| [44] | Low control priming[1] | Experimental | Between-groups | No | Belief in Conspiracy Theories Inventory (BCTI Swami et al., 2010, 2011). | Specific | 0.88 | 0.02 |
| [26] | Ostracism priming[1] | Experimental | Between-groups | No | Conspiracy beliefs of 14 political events | Specific | 0.02* | NR |
| [26] | Pain priming[1] | Experimental | Between-groups | No | Conspiracy beliefs of 14 political events | Specific | 0.78 | NR |
| [26] | Ostracism priming[1] | Experimental | Between-groups | No | Conspiracy beliefs of 6 political events | Specific | 0.03* | NR |
| [26] | Ostracism priming/self-affirmation[3] | Experimental | Between-groups | No | 10-item version of the Generic Conspiracist Beliefs scale (GCBS; Brotherton, French, & Pickering, 2013), | General | 0.001** | NR |
| [26] | Ostracism priming/no affirmation[1] | Experimental | Between-groups | No | 10-item version of the Generic Conspiracist Beliefs scale (GCBS; Brotherton, French, & Pickering, 2013), | General | 0.89 | NR |
| [45] | Narrative persuasion | Experimental | Between-groups | Unclear | Self-made 2-item measure | General | 0.086 | NR |
| [45] | Narrative persuasion | Experimental | Between-groups | Unclear | Conspiracist Mentality Questionnaire (Lantian, Muller, Nurra, & Douglas, 2016). | General | 0.47 | NR |
| [46] | Pseudoscience class | Experimental | Within-groups | Unclear | Conspiracy belief subscale of the Paranormal Beliefs Scale (Tobacyk, 2004). | Specific | < 0.0001** | 1.07 |

(*Continued*)

**Table 1.** (Continued)

| Author | Intervention | Design | Grouping | Aware of Intervertion | Conspiracy measure | Type of Measure | p-value | Cohen's d |
|---|---|---|---|---|---|---|---|---|
| [46] | Research Methods class | Experimental | Within-groups | Unclear | Conspiracy belief subscale of the Paranormal Beliefs Scale (Tobacyk, 2004). | Specific | 0.14 | NR |

NR = Not Reported; Design = the experimental design used in the intervention; Experimental = the study used two independent groups; Quasi-experimental = the study used the same participants for both conditions; Aware of Intervention = whether participants were aware of the true nature of the experiment; Yes = participants were aware of the interventions true intentions; No = the true nature of the interventions was not revealed to participants; Unclear = the authors do not establish whether the participants were aware or unaware of the intervention.; Type of Measure = whether the conspiracy measure used measured broad conspiracy ideation, or examined a specific set of beliefs; General = the scale used measured general conspiracy ideation; Specific = the scale measured belief a specific set of conspiracy theories; * indicates $p < 0.05$; ** indicates $p < 0.01$.[1] = these interventions were designed to increase conspiracy beliefs, as a comparison for the other interventions. [2] = these metainoculations were designed to nullify the effects of the inoculation interventions used in prior studies, not to reduce conspiracy beliefs. [3] = these self-affirmations were designed to nullify the effects of ostracism on increasing conspiracy beliefs.

neutral groups. However, only a small number of these interventions demonstrated large effect sizes ($k = 3$). Furthermore, medium effects were only found with a handful of interventions ($k = 2$). The remaining studies were found to have only small, or very small effects.

## Interventions aimed at reducing conspiracy beliefs

The effect sizes of interventions aimed at reducing conspiracy beliefs are reported in Fig 2. Four interventions aimed at reducing conspiracy beliefs were omitted from the graph as they only reported statistical significance, and the Cohen's d could not be calculated. These interventions were a single conspiracy labelling intervention ($k = 1$), narrative persuasion interventions ($k = 2$), and a research methods module ($k = 1$). This section also omits any interventions that aimed to increase conspiracy beliefs.

Approximately half of the examined interventions consisted of *priming* based tasks ($k = 15$). The majority of these interventions demonstrated a significant change in conspiracy beliefs ($k = 10$). These effects were all either small or very small. Participants who were primed to be less susceptible to persuasion tactics showed significantly lower conspiracy beliefs when

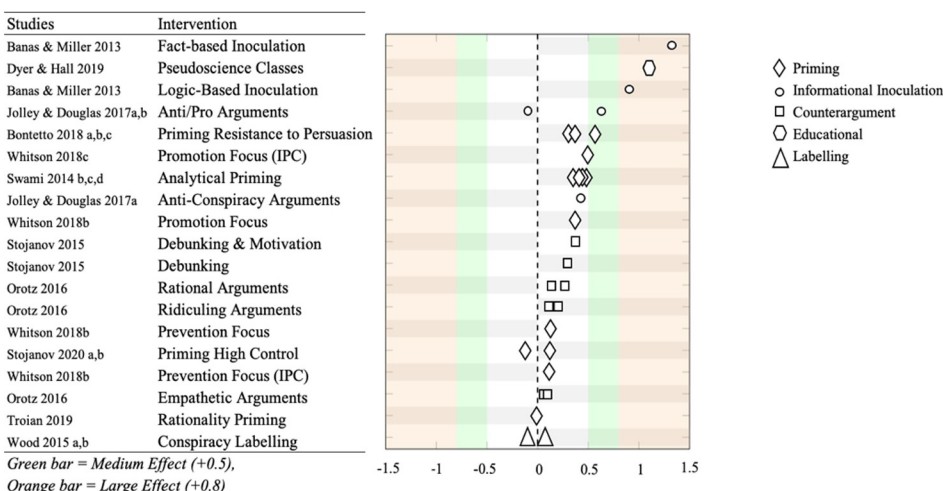

**Fig 2. Cohen's d effect sizes interventions aimed at reducing conspiracy beliefs.** *Note*: This figure has been adapted from [51].

compared to controls among three experimental comparisons. These effects were shown to range from small to medium ($d$ = 0.3–0.56). Interventions that primed participants to engage in analytical thinking resulted in primed participants having lower conspiracy belief than controls ($k$ = 4). However, the effects of these differences were small. Other priming interventions focused on manipulating participants' sense of control. Priming participants to feel like they had a higher sense of control had mixed results, either increasing or decreasing conspiracy beliefs with very small effects ($d$ = -0.13–0.11) [44].

About a sixth of interventions in the analysis used inoculation methods ($k$ = 5). All were successful at reducing conspiracy beliefs, relative to controls, all with either medium or large effects. Inoculations that identified the factual inaccuracies of conspiracy beliefs were found to be the most effective of all the interventions in the review ($d$ = 1.3) [21]. Jolley and Douglas [15] only reported medium effects for inoculation methods in terms of reducing conspiracy belief measures. Inoculations that demonstrated the logical fallacies of conspiracy beliefs were found to be the second most effective intervention ($d$ = 0.9) [21].

In contrast to inoculations, traditional counterarguments only produced small and very small effects. Rational counterarguments that described the factual inaccuracies of conspiracy theories were found to have only very small to small effects ($d$ = 0.13–0.27) [42]. Stojanov [16], reported similar results, with counterarguments only producing small effects in terms of changing conspiracy beliefs ($d$ = 0.29). These counterarguments were found to be slightly more effective, when they included an extra component that described to participants both the motives of conspiracy theories, and the logical fallacies in the conspiratorial thinking ($d$ = 0.37). Counterarguments that appealed to participant's sense of empathy, outlining the damages that can result from conspiracy beliefs was found to have very small effects ($d$ = 0.05–0.1), thus being one of the least effective interventions in the sample [42]. Finally, counterarguments that attempted to ridicule those who held conspiracy beliefs also produced only very small effects in terms of reducing conspiracy beliefs ($d$ = 0.11–0.2) [42].

We also found a number of standalone measures that utilised novel interventions to confront conspiracy belief. One study used an in-person 3-month educational course that aimed at explicitly teaching students how to differentiate between proper scientific and pseudoscientific practices. At the end of the term, the pseudoscience class reported significantly lower conspiracy belief in comparison to a typical university research methods class [46]. The effect was also found to be large ($d$ = 1.07). Furthermore, this was one of the few studies that was within-subjects. It was the sole study to use a longitudinal design (3-month interval between pre and post measures). Another study found that labelling conspiracy statements as "conspiracy theories" in comparison to "ideas", or "political scandals", had no effect in changing belief in conspiracy theories ($p$ = >0.05, $d$ = -0.1–0.07) [43].

A handful of studies reported no Cohen's d effects, nor provided sufficient information to calculate them. As per the protocol, in this circumstance we reported the statistical significance of these interventions. One study investigating the effects of narrative persuasion on conspiracy belief was found to have no significant effects [45].

Finally, our analysis revealed a number of "backfiring" interventions; those that aimed at reducing conspiracy belief but paradoxically increased conspiracy beliefs. While labeling conspiracy theories did not produce any significant effects, one study found that it had a small effect in increasing conspiracy beliefs ($d$ = -0.1). Likewise, priming participants to feel like they had a higher sense of control increased conspiracy beliefs in one study ($d$ = -0.13). Anti-conspiracy inoculations followed by pro-conspiracy arguments, while one of the most effective interventions found in this review at reducing conspiracy beliefs, also increased conspiracy beliefs in one study albeit with very a small effect ($d$ = -0.1).

**Interventions aimed at increasing conspiracy beliefs..** In addition to the interventions shown in Fig 2, which were all designed to reduce conspiracy belief, the review also found a number of interventions that were not intended to reduce conspiracy belief (and are therefore not shown in Fig 2). Most of these interventions were used as a comparison to interventions within the same study intending to reduce conspiracy beliefs. One study used informational inoculations, where participants were presented with arguments supporting conspiracy beliefs before they were shown conspiratorial content [15]. This interventions was found to have a medium effect in increasing conspiracy beliefs ($d$ = -0.66). Within the same article, two interventions similarly used pro-conspiracy inoculations but were then followed by anti-conspiracy arguments to test whether the order in which these inoculation were presented would influence their effects. The results for these two interventions were mixed ranging from a small effect in increasing conspiracy beliefs and a small effect in decreasing conspiracy beliefs ($d$ = -0.39–0.31). Another study primed participants to feel lower levels of perceived control with the intention of increasing conspiracy beliefs ($k$ = 4). Two low control priming interventions were found to have very small to very small effects in increasing conspiracy beliefs ($d$ = -0.13 –-0.15). However, the remaining two low control priming interventions had small to very small effects in decreasing conspiracy beliefs ($d$ = 0.02–0.27). Finally, one study tested the effects of meta-inoculations, where researchers told participants what inoculations were, with the intention of investigating whether the initial effects of informational inoculations could be negated. Meta-inoculations were another form of inoculations that were found to be reasonably effective in changing conspiracy beliefs, with medium effects, and thus were able to moderately negate the effects of anti-conspiracy inoculations ($d$ = 0.58). Another study found that when participants were primed to feel ostracised, they were found to be more likely to hold conspiracy beliefs than control groups ($p$ = 0.02) [26]. However, it was also reported that when participants who were in the ostracism condition were instructed to think of values that were important to them, this was shown to significantly negate the effects of the ostracism priming [26]. The study reported no effect sizes for these interventions.

## Discussion

The aim of this review was to assess how researchers are currently developing interventions to reduce susceptibility to conspiracy beliefs. We found that only half of the interventions examined induced any change in the degree in which participants agreed with conspiracy statements. Furthermore, only a handful of interventions produced medium to large effects. These findings suggest that most existing conspiracy interventions are ineffective in terms of changing conspiracy beliefs. This finding is in line with previous research that suggests that conspiratorial thinking may be underpinned by inherent susceptibilities [52] and may not be easily refuted with standard counterarguments [13,17,19].

The few successful interventions in our sample shared a number of common characteristics. The majority of these interventions consisted of treatments that were given *before* participants were exposed to conspiracy statements. The first category consisted informational inoculations, of which the two interventions with the largest observed effect sizes belong. These findings suggest that the most effective conspiracy interventions are those take place before participants have been exposed to conspiracy beliefs. Pre-emptively refuting the inaccuracies of conspiracy beliefs was much more effective than traditional counterarguments that were given after participants had encountered conspiracy theories. Traditional counterarguments and debunking strategies in comparison only produced very small effects, whereas inoculation strategies ranged from medium to large effects. However, one pitfall of inoculation-based interventions is that their ability to reduce conspiracy belief was easily counteracted. For

participants who were warned about the inoculation interventions, and told there would be an attempt to change their conspiracy beliefs, the effects of the inoculation treatment were rendered negligible [21]. These warnings were referred to as *metainoculations*. All that was required to make these interventions useless was to warn participants about them beforehand. In essence, the effectiveness of inoculation strategies can be used to counteract their own effects. The effects of these meta-inoculations were found to be in the medium range.

The second category of successful interventions consisted of priming interventions, particularly those that attempted to instill critical thinking skills in their participants. Swami et al [25], found that by having participants evaluate pieces text in fonts that were difficult to read, they were more likely to engage in analytical thinking. Other research has shown that participants that were primed to feel more in control had lower conspiracy beliefs when compared to the baseline group. Recent research has also found that participants who were primed to have a higher resistance to persuasion were shown to be less susceptible to conspiracy beliefs than those in control group [27]. These findings are noteworthy when considering recent research regarding how people engage with conspiracy theories. Previous research has indicated that conspiracy beliefs are the product of intuitive, emotional thinking and that deliberation and focused analysis is associated with lower susceptibility to conspiracy beliefs [53,54]. As such, fostering critical and analytical thinking in participants through priming might possibility motivate them to push past superficial evaluations, and examine the content of conspiracy beliefs. While the possibility of priming members of the public to engage with conspiracy beliefs more critically may be promising, the feasibility of such a priming intervention outside a controlled lab environment has yet to be ascertained. Whether one priming session would be sufficient to foster critical evaluation of conspiracy theories has not yet been demonstrated. It should also be noted that all the analytical priming interventions included in this review originate from one 4-paper study [25]. More independent research may be needed to replicate the results of this intervention. Furthermore, there are some drawbacks associated with the manipulation checks used during these interventions. Participants' susceptibility to the Moses Illusion was used as a manipulation check to assess whether reading difficult-to-read fonts successfully elicited analytical processing. However, previous research has failed to replicate whether reading in a difficult font has a significant effect on susceptibility to the Moses Illusion [55].

Among the interventions which reported medium-to-large effects, there was one intervention which was characteristically different to the rest. This intervention made use of a university module to teach students about the differentiation between good scientific practice and pseudoscience [46]. After taking the module, students endorsed far fewer conspiracy theories. This was the sole study in this review with a large effect size that did not administer the intervention before participants were exposed to conspiracy statements.

It should be noted that the three intervention types that were most effective within this review are difficult to administer in a real-world setting. Informational inoculations need to be administered to participants before they encounter a conspiracy which would be difficult to maintain in practice. Priming interventions are difficult to maintain as people encounter a variety of new sources every day. Finally, educational interventions require a substantial amount of effort and commitment from both the educator and participants. None of these interventions will provide and easy solution to conspiracy beliefs practically speaking. As such, future research should examine whether the mechanisms of long-form interventions such the three-month university pseudoscience module [46], can be condensed into a short-form counterpart. Research in this area may help bridge the gap between effective but unscalable interventions and interventions that can realistically be implemented in everyday life. In order to efficiently counter misinformation spread by conspiracy beliefs, future

interventions will need to focus on developing a solution that will be more easily implemented.

## Limitations of studies examined in the review

There were a number of limitations found in the studies examined in this review. The main limitations identified are as follows: focusing only on one form of research design, lack of cross-culture sampling, and issues with the measurement of conspiracy beliefs. Firstly, one of the potential limitations of the findings of this review is the approach taken to research design in the studies included in our sample. The majority of the studies in this sample used cross-sectional designs. Where within-groups comparisons were used, it was over a relatively small interval. As such, it cannot be ascertained whether the beneficial effects of the interventions found in this study will remain consistent after a period of time. The interventions only examined short term effects. Therefore, it is difficult to conclude whether even the most effective intervention found in this review will be of much use in tackling conspiracy beliefs outside the lab. This limitation also applies to interventions that intended to temporarily alter psychological states of participants (e.g., sense of control). As these studies aim at only temporarily changing the psychological states of their participants, it's to be expected that any effects observed would only last until the effects of experimental manipulation wore off. Furthermore, as identified by Swire-Thompson et al [56], one of the main issues with cross-sectional studies that exclusively use post-intervention measures of belief is that there is no way of establishing whether the groups tested were matched at baseline. The conspiracy beliefs of either the control group or one of the experimental conditions may already have been larger than the other groups before the intervention.

The second limitation we identified is the lack of cultural diversity within the samples tested. Ideally, studies that wish to identify conspiracy interventions that can be widely used should use cross-cultural studies to verify their effects are only observed a specific cultural context. The studies in this review were predominantly conducted with western samples, with the majority of studies coming from either North America or the United Kingdom. While some studies, such as Poon et al [26], were conducted outside of the West, many of these studies recruited American samples using Amazon's Mechanical Turk. It should also be noted that the 24 studies included in this review only represent 10 groups of authors. Furthermore, half of these studies originate from three research groups, which may suggest a possible source of bias.

Finally, many of the studies used novel, custom measures of conspiracy ideation to measure conspiracy belief. The lack of consistency in measurement makes it difficult to generalize positive findings of these studies. Many studies only examined belief in specific conspiracy theories. Such findings can be difficult to generalize, as an intervention that is successful in changing anti-vaccine conspiracy beliefs may not be particularly effective when used to confront 9/11 conspiracy beliefs. Furthermore, a number of the conspiracy belief measures that were used within these studies were also self-designed, even when general conspiracy belief was being measured. Finally, exclusively using measures of conspiracy belief makes it difficult to draw any conclusions regarding how these interventions affect the underlying cognitive process behind conspiracy beliefs. A focus on more consistent usage of valid and reliable conspiracy measures would be desirable in future studies.

## Limitations of the current systematic review

There were also some possible limitations in the systematic review itself. The outcome of interest for this review was the Cohen's d effect size, meaning that this review focused

mainly on pairwise comparisons. While this allowed for consistency when comparisons were made between the interventions, it also came with some drawbacks. For example, a number of studies examined additional variables that may have mediated the changes induced by the interventions. However, as this review focused of pairwise comparisons, some of these details may not have been examined.

## Implications for research / practice / design

This review identified a number of directions for future research. Firstly, we recommend that future studies on conspiracy belief interventions focus on using longitudinal designs. Understanding the effects of these interventions beyond their immediate changes is important in terms of identifying how effective they are at challenging conspiracy beliefs. Second, we recommend that future studies be conducted with non-western populations, with more of a focus on cross-cultural sampling. Conspiracy beliefs may often be specific to cultural context, as such we recommend that research be expanded beyond British and American samples.

In terms of practical implications for challenging conspiracy beliefs, we recommend that those with an interest in reducing the misinformation that conspiracy theories spread should do the following:

1. *Avoid appealing to emotions and affect*: Interventions that manipulated the emotional state of participants, or appealed to feelings of empathy had small effects in terms of changing conspiracy beliefs.

2. *Counterarguments are not effective*: Counterarguments against specific conspiracy beliefs that are given after participants have been exposed to a conspiracy theory tend not to be particularly effective.

3. *Prevention is the best cure*: Interventions that provided counterarguments for conspiracy theories were most effective when the counterargument came before the participants were exposed to the particular conspiracy theories that the study focused on. The findings suggest it is more difficult to challenge conspiracy beliefs once participants have started to believe in them. If participants have been taught why certain conspiracy theories are implausible before they have been exposed to conspiratorial media they are much more resistant to conspiracy beliefs.

4. *An analytical mindset and critical thinking skills are the most effective means of challenging conspiracy beliefs*: Participants who were primed to have an analytical mindset were less likely to have conspiracy beliefs than controls. Furthermore, when interventions moved beyond putting participants in an analytical mindset, and actually explicitly taught them how to evaluate conspiracy beliefs using specific critical thinking skills, they were much less likely to have conspiracy beliefs.

## Conclusion

In conclusion, this review sought to investigate how effective current interventions are at changing conspiracy beliefs. We found that overall, the majority of current conspiracy interventions are ineffective in terms of changing conspiracy beliefs. Despite this, we have identified several promising interventions that may be fruitful to pursue in future studies. We propose that a focus on inoculation-based and critical thinking interventions will bear

more promising results for future research, though further efforts are needed to reduce participant burden and more easily implement these interventions in the real world.

## Supporting information

**S1 Table. Demographic and publication details of studies included in the review.**
(DOCX)

**S1 Fig. Risk of bias assessment.**
(TIF)

**S1 File. PRISMA checklist.**
(DOCX)

## Acknowledgments

We thank Dr. Rebecca Umbach (Google) for comments on the manuscript. We thank Dr. Joe Campbell (University of Nottingham) for providing his LaTeX script to help develop Fig 1 in this manuscript.

## Author Contributions

**Conceptualization:** Gillian Murphy, Conor Linehan.

**Methodology:** Cian O'Mahony, Maryanne Brassil, Gillian Murphy, Conor Linehan.

**Supervision:** Conor Linehan.

**Visualization:** Cian O'Mahony.

**Writing – original draft:** Cian O'Mahony.

**Writing – review & editing:** Cian O'Mahony, Maryanne Brassil, Gillian Murphy, Conor Linehan.

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
