## [Decision Letter · Decision Letter 0]

2 Nov 2022

PONE-D-22-25039The efficacy of interventions in reducing belief in conspiracy theories: a systematic reviewPLOS ONE

Dear Dr. O'Mahony,

Thank you for submitting your manuscript to PLOS ONE. After careful consideration, we feel that it has merit but does not fully meet PLOS ONE’s publication criteria as it currently stands. Therefore, we invite you to submit a revised version of the manuscript that addresses the points raised during the review process.

Your paper has been revised by four expert reviewers. They highlighted significant changes to be made to improve your paper. You are therefore invited to make the best use of those suggestions to provide a revised version of your paper.

We look forward to receiving your revised manuscript.

Kind regards,

Pierluigi Vellucci

Academic Editor

PLOS ONE

Journal Requirements:

2. Please amend either the abstract on the online submission form (via Edit Submission) or the abstract in the manuscript so that they are identical.

4. We note you have included a table to which you do not refer in the text of your manuscript. Please ensure that you refer to Table 2 in your text; if accepted, production will need this reference to link the reader to the Table.

5. Please include your tables as part of your main manuscript and remove the individual files. Please note that supplementary tables (should remain/ be uploaded) as separate "supporting information" files.

Reviewers' comments:

Reviewer's Responses to Questions

**Comments to the Author**

1. Is the manuscript technically sound, and do the data support the conclusions?

Reviewer #1: Yes

Reviewer #2: Yes

Reviewer #3: Yes

Reviewer #4: Partly

2. Has the statistical analysis been performed appropriately and rigorously? 

Reviewer #1: I Don't Know

Reviewer #2: Yes

Reviewer #3: Yes

Reviewer #4: N/A

3. Have the authors made all data underlying the findings in their manuscript fully available?

Reviewer #1: Yes

Reviewer #2: Yes

Reviewer #3: Yes

Reviewer #4: Yes

4. Is the manuscript presented in an intelligible fashion and written in standard English?

Reviewer #1: Yes

Reviewer #2: Yes

Reviewer #3: Yes

Reviewer #4: Yes

5. Review Comments to the Author

Reviewer #1: This is an interesting paper on an increasingly popular topic of study: conspiratorial though (CT). While this area of research has been studied extensively, especially in the time of COVID and Trump, the current study is a novel meta-analysis of CT interventions. While most papers look at the causes and consequences of CT, as shown in the paper far fewer test methods to combat this style of thinking. I have comments below I'd like to see addressed, but overall this seems like a useful contribution to this ever-growing area of research.

-Abstract/Discussion: The premise that CT is on the rise is not accurate. Please see Uscinski et al. (2022a) for a test of this hypothesis. This is not a fatal flaw because, as the authors articulate well, there are many negative externalities associated with CT (regardless of whether it is on the increase). That said, I would not frame the paper around the assumption that CT is having a renaissance.

-Pages 6, 9: For readers who do not conduct meta analyses themselves, please define PRISMA more clearly. Please apply this comment to any other meta-analysis terminology that might not be familiar to a lay reader.

-Page 6: Why was Google Scholar not included in the search?

-Table 1: Given the amount of information here, can one perform a multivariate analysis? For example, with Cohen's d as the DV, and the other columns in the table as IVs?

-Page 20/Figure 2: Is it possible to add CIs to the Cohen's d plots? I would also sort this figure ascending, rather than by intervention type, to group the most/least effective interventions together. This will tie more directly into the discussion of patterns in the results on pp. 13-22.

-Pages 22ff: A key takeaway of the paper is that CT is a deeply-embedded psychological characteristic that is hard to change. I would discuss this a bit more. Please see work by Uscinski et al. on this (2022a,b; 2021; 2016).

-Pages 25-26: In the discussion of limitations I would note that the 24 studies represent only 10 groups of authors. Moreover, half of the studies come from just three groups. I am mindful that the analysis is limited by what research has been published, but were I writing this paper I would point this out as a possible source of bias.

REFERENCES

Uscinski, JE et al. 2022a Have beliefs in conspiracy theories increased over time? PLoS ONE 1 7(7): e0270429.

Uscinski, JE, et al. 2022b Cause and Effect: On the Antecedents and Consequences of Conspiracy Theory Beliefs. Current Opinion in Psychology 47: 101364.

Uscinski, JE, et al. 2021 Do Conspiracy Beliefs form a Belief System?: Examining the Structure and Organization of Conspiracy Beliefs. Journal of Social and Political Psychology 9: 255-271.

Uscinski, JE, et al. 2016 What Drives Conspiratorial Beliefs? The Role of Informational Cues and Predispositions. Political Research Quarterly 69: 57-71.

Reviewer #2: There is a lot to like in this paper. The review of this kind is timely and much needed. The authors do a good job motivating this study - they build a persuasive case.

I also liked the categorization of the interventions they propose and the way it was argued.

The fact that they pre registered the study and shared materials and data made it much easier to evaluate and use by future researchers (the osf documentation is clearly labeled and easy to follow) - much appreciated.

The authors closely followed PRISMA protocol.

They were careful not to overclaim in the Discussion section and to rely exclusively on their results.

That being said, I have a few major suggestions:

First, I would like the authors to acknowledge the fact that priming interventions are in fact a very heterogeneous group - explicitly state what are their shared features, but also how they differ amongst themselves.

Second, the effect sizes of interventions in both directions (aimed to decrease but also to increase CT beliefs, e.g. Jolley & Douglas, 2017a) are lumped together. I would suggest redoing the analysis omitting the interventions increasing CT beliefs and also commenting the effect sizes for the two types of interventions.

Next, the authors should make sure to address the fact that the analytical thinking priming interventions stem from one 4-study paper (Swami et al., 2014). Before implementing these types of interventions widely, it would be good to test the robustness of the effect in an independent replication study. I would also like the authors to discuss the upsides and drawbacks techniques for experimental priming of analytical thinking in this research (e.g. reading in a difficult font) and its manipulation checks (e.g. susceptibility to Moses illusion). The duration of the priming effects is also unknown.

I also have a few minor things to suggest:

First, it would be better to avoid over simplified explanations, for example:

"The tendency to believe in contradicting conspiracy The efficacy of interventions in reducing belief conspiracy theories theories suggest that those who believe in conspiracies tend to make superficial judgements about whether they are true."

Please consult these papers about contradictory conspiracy theories (they discuss whether they necessarily reflect superficial judgments about the veracity of the CTs):

Lukić, P., Žeželj, I., & Stanković, B. (2019). How (ir) rational is it to believe in contradictory conspiracy theories?. Europe’s journal of psychology, 15(1), 94-107.

Wood M. J. (2017). Conspiracy suspicions as a proxy for beliefs in conspiracy theories: Implications for theory and measurement. British Journal of Psychology, 108(3), 507–527.

Second, I would also suggest rewording emotionally charged claims such as:

By synthesizing the evidence, we may better prepare for future real-world threats from conspiracists.

Reviewer #3: This is a good paper. I think some rewriting would make it publishable.

1. There is no "striking increase in conspiracy theory beliefs." You don't need to make this throwaway claim.

See: J. Uscinski et al., Have beliefs in conspriacy theories increased over time? Plos One 17, e0270429 (2022).

D. Romer, K. H. Jamieson, Conspiracy theories as barriers to controlling the spread of COVID-19 in the US. Social Science & Medicine 263, 1-8 (2020).

M. Mancosu, S. Vassallo, The life cycle of conspiracy theories: evidence from a long-term panel survey on conspiracy beliefs in Italy. Italian Political Science Review/Rivista Italiana di Scienza Politica 52, 1-17 (2022).

2. I am not sure what the abstract means when it says that little research has reviewed methods for reducing beliefs. I have seen tons of studies on this topic. And lots of critiques of those studies.

3. Use newer citations for the intro. Many of the cites are more than five years old. No need for that.

4. Grimes is not a good citation for conspiracy beliefs being resistant to refutation. He doesn't test that.

5. Claiming that conspiracy theory beliefs have harmful outcomes assumes causality. I think such assumptions are unwarranted at this time.

6. The intro should almost be entirely removed - the paper can stand on its own without a broad review on conspiracy theories - just focus on the interventions, that's it. Don't cite so much stuff either, just cite the Douglas et al 2019 lit review, for example.

7. pg.6. Not sure that that conspiracy theory beliefs have "very real public health threats." I think the jury is still out. See:

J. Uscinski, A. M. Enders, C. Klofstad, J. Stoler, Cause and Effect: On the Antecedents and Consequences of Conspiracy Theory Beliefs. Current Opinion in Psychology https://doi.org/10.1016/j.copsyc.2022.101364, 101364 (2022).

8. Did the search miss "conspiratorial" because it has a t instead of a c at the end? How about misperceptions and rumors?

9. There needs to be a better demarcation between general conspiracy mentality measures and specific measures of conspiracy theory beliefs. These are different things, and the implications of decreasing one versus the other are very different.

10. Figure 2 is hard to read. The dots are too far from the labels.

11. page 22: Conspiracy theories are not "becoming" widespread, and the literature cited, Oliver and Wood and van der linden show no such thing. Also those cites are both really old for making claims about recent over time change.

12. Your paper is about conspiracy theories, not conspiracies. Be sure to use the term conspiracy theories, for example, on bottom of page 23 but elsewhere too. Conspiracies are real.

13. Page 27. What is a "misinformation effect of conspiracy theories"? This is clunky language.

14. The paper, in general, needs to focus on the meta analysis, and the findings of that. Trim the front substantially and stay laser focused on the basic findings. This will highlight the paper's value. I think the abstract could highlight the main findings more: what works and what doesn't? In other words, only focus on interventions and nothing else.

Reviewer #4: This paper is a systematic review of the efficacy of interventions in countering belief in conspiracy theories. It is a very meaningful study as the field is growing fast and a review is warranted. However, the paper could benefit from some substantial changes.

Major comments:

- Because the search was conducted in June 2021, the results feel a little outdated, as with COVID-19 there is a burgeoning number of studies that look at the issue in an empirical setting, which are not included. It may be too much to ask the authors to conduct another updated search, considering the amount of anticipated work associated with it. That said, if the authors intend to keep the included papers as they are, maybe a more in-depth discussion is necessary regarding the implications for future research given all the new emerging studies.

- Relatedly, the included studies are not a lot, even though each paper investigates several interventions. The authors mentioned that many of the studies use misinformation, fake news, and conspiracy theories interchangeably. Then why not include all of them in the keyword search to expand the sample? Or at least, justify why only conspirac* is used in the search strategy.

- In the background section, besides explaining the different types of interventions, it's also helpful to include a discussion on the different cognitive processes of conspiracy beliefs - proportionality bias, intentionality bias, pattern perception, jumping to conclusions, confirmation bias, and the conjunction fallacy. And if possible, relate the interventions to these categories.

- Is there any systematic difference between the more general and more specific conspiracy beliefs?

- As discussed in the limitations section, the paper lacks external validity. What kind of potential interventions in real life can we devise, for example, on social media platforms?

Minor comments:

- The PRISMA flowchart can be more detailed - breaking down the number of papers excluded according to the reason/criteria

- Period missing at the end of page 30.

6. PLOS authors have the option to publish the peer review history of their article (what does this mean?). If published, this will include your full peer review and any attached files.

Reviewer #1: No

Reviewer #2: No

Reviewer #3: No

Reviewer #4: No

---

## [Author Response · Author response to Decision Letter 0]

21 Dec 2022

Authors: We wish to sincerely thank the reviewers for taking the time to carefully consider our paper and for the suggestions you have all contributed to this manuscript. Below we have included our responses and we hope we have adequately addressed the issues you have raised. In the revised manuscript, the changes are visible in blue font. 

Reviewer 1

-Abstract/Discussion: The premise that CT is on the rise is not accurate. Please see Uscinski et al. (2022a) for a test of this hypothesis. This is not a fatal flaw because, as the authors articulate well, there are many negative externalities associated with CT (regardless of whether it is on the increase). That said, I would not frame the paper around the assumption that CT is having a renaissance.

Authors: Thank you for raising this point. We became aware of the cited study shortly after our submission and agree that statement is not accurate and so have removed it from the abstract

-Pages 6, 9: For readers who do not conduct meta analyses themselves, please define PRISMA more clearly. Please apply this comment to any other meta-analysis terminology that might not be familiar to a lay reader.

Authors: Thank you for the suggestion. We have provided a brief explanation of the PRISMA process. We also refer readers to our PRISMA checklist in supplementary materials as a more complete summary of what the 27 items on the checklist entail on page 6 of the manuscript.

-Page 6: Why was Google Scholar not included in the search?

Authors: Based on our preparatory work, we agreed that the three databases we searched were sufficient in identifying these papers. As Google Scholar primarily crawls other databases and identify the same articles that our existing databases would already find. As such, we did not see the need to include it in our searches – this is typical for systematic reviews, where Google Scholar is not usually included. 

-Table 1: Given the amount of information here, can one perform a multivariate analysis? For example, with Cohen's d as the DV, and the other columns in the table as IVs?

Authors: This is an interesting suggestion. Unfortunately, as identified by another reviewer, the interventions and their measures are quite heterogenous, meaning any multivariate analysis would not provide any useful results. As such, we concluded that we would confine the analysis of this paper to the narrative synthesis.

-Page 20/Figure 2: Is it possible to add CIs to the Cohen's d plots? I would also sort this figure ascending, rather than by intervention type, to group the most/least effective interventions together. This will tie more directly into the discussion of patterns in the results on pp. 13-22.

Authors: Thank you for the suggestion. With regards to adding CIs to the plots, there are two main problems. First, due to the fact that we include multiple effect sizes in each row it would be difficult to see the overlapping CI stems. Secondly, the manner of which the Cohen’s d was reported each study differed greatly. As such, some papers would not have sufficient data to calculate the CIs.

Regarding your suggestion to sort the data from most effective to least, we have rearranged the figure according to your suggestion. This alteration also addresses another comment made by another reviewer, where the dots are too far from the interventions/authors. This change should place the data points closer to those labels.

-Pages 22ff: A key takeaway of the paper is that CT is a deeply-embedded psychological characteristic that is hard to change. I would discuss this a bit more. Please see work by Uscinski et al. on this (2022a,b; 2021; 2016).

Authors: Thank you for recommending these citations. We have included a brief discussion regarding this topic on page 24. 

-Pages 25-26: In the discussion of limitations I would note that the 24 studies represent only 10 groups of authors. Moreover, half of the studies come from just three groups. I am mindful that the analysis is limited by what research has been published, but were I writing this paper I would point this out as a possible source of bias.

Authors: Thank you for highlighting this – we have mentioned this within our limitations section as a potential source of bias on page 29.

Thank you for taking the time to review our work, we hope you find the revised manuscript suitable for publication. 

 

Reviewer 2

First, I would like the authors to acknowledge the fact that priming interventions are in fact a very heterogeneous group - explicitly state what are their shared features, but also how they differ amongst themselves.

Authors: Thank you for pointing this out – while it was useful to group interventions into broader categories for the sake of our analysis and communicating our findings, we agree that priming interventions in particular are quite a heterogenous group. We have made note of this in our manuscript, on page. 5.

Second, the effect sizes of interventions in both directions (aimed to decrease but also to increase CT beliefs, e.g. Jolley & Douglas, 2017a) are lumped together. I would suggest redoing the analysis omitting the interventions increasing CT beliefs and also commenting the effect sizes for the two types of interventions.

Authors: Thank you for this suggestion. We agree that the interventions that aimed to reduce conspiracy beliefs are the primary target of this review and it is important to clarify the effects of these interventions, separate from those used as comparators that were not expected to reduce conspiracy beliefs. We have made two amendments to make this clearer to readers. First, we provide a section on pages 22-24 within the results section that discusses the effects of interventions that both aimed to increase conspiracy belief, and those that aimed to reduce belief but paradoxically increased it. Secondly, we have removed any interventions that aimed to increase conspiracy beliefs from Figure 2 to make the results clearer.

Next, the authors should make sure to address the fact that the analytical thinking priming interventions stem from one 4-study paper (Swami et al., 2014). Before implementing these types of interventions widely, it would be good to test the robustness of the effect in an independent replication study. I would also like the authors to discuss the upsides and drawbacks techniques for experimental priming of analytical thinking in this research (e.g. reading in a difficult font) and its manipulation checks (e.g. susceptibility to Moses illusion). The duration of the priming effects is also unknown.

Authors: Thank you for identifying this – we have noted in the discussion section (page 26) that the analytical priming interventions originate from one paper and we draw attention to the need for independent replication. We have also critiqued some of the techniques used in this study (i.e. difficult to read font) as other studies indicate that the manipulation checks in question (such as susceptibility to the Moses illusion) fail to replicate.

I also have a few minor things to suggest:

First, it would be better to avoid over simplified explanations, for example:

"The tendency to believe in contradicting conspiracy. The efficacy of interventions in reducing belief conspiracy theories suggest that those who believe in conspiracies tend to make superficial judgements about whether they are true."

Please consult these papers about contradictory conspiracy theories (they discuss whether they necessarily reflect superficial judgments about the veracity of the CTs):

Authors: Thank you for referring us to these studies – in light of these findings we have removed the quoted statement from our manuscript on page 25. We have chosen to retain the message of that paragraph, as other research has suggested that deliberation and System 2 thinking has been associated with fewer conspiracy beliefs/ reduced conspiratorial thinking (Bago et al., 2022; Forgas & Baumeister, 2019). As such, we still draw the conclusion that analytical thinking may help individuals consider information more clearly (these changes can be seen on page 25). However, we have removed contradicting conspiracy beliefs as supporting evidence for this suggestion.

Second, I would also suggest rewording emotionally charged claims such as:

By synthesizing the evidence, we may better prepare for future real-world threats from conspiracists.

Authors: Agreed – we have removed this sentence from page 6 of the manuscript.

Thank you for your constructive comments, we hope you now find the manuscript suitable for publication. 

 

Reviewer #3: 

This is a good paper. I think some rewriting would make it publishable.

1. There is no "striking increase in conspiracy theory beliefs." You don't need to make this throwaway claim.

Authors: Thank you for highlighting this – we have removed this statement and similar statements from the abstract and page 25 at the beginning of the discussion section.

2. I am not sure what the abstract means when it says that little research has reviewed methods for reducing beliefs. I have seen tons of studies on this topic. And lots of critiques of those studies.

Authors: Thank you for highlighting this – we have reworded the abstract. Our initial intention was to indicate that a systematic review of such interventions in a broad context had yet to be conducted. We agree that the initial wording makes it read as if we are suggesting little research has been conducted on conspiracy belief interventions themselves. We have revised this statement to better communicate our position.

3. Use newer citations for the intro. Many of the cites are more than five years old. No need for that.

Authors: Thank you for the suggestion – we have included a number of more recent citations in our introduction section, for example; (Bierwiaczonek et al., 2022; Imhoff et al., 2022; Knobel et al., 2022; Mulukom et al., 2020; Pytlik et al., 2020; Sutton & Douglas, 2020)

4. Grimes is not a good citation for conspiracy beliefs being resistant to refutation. He doesn't test that.

 Authors: Thank you for highlighting this – we have removed this section from page 3 as part of the streamlining of the introduction. 

5. Claiming that conspiracy theory beliefs have harmful outcomes assumes causality. I think such assumptions are unwarranted at this time.

Authors: Thank you for highlighting this – we have indicated in the manuscript on page 3 that many of these findings are demonstrating associations between these phenomena and causal conclusions cannot be made at the current time.

6. The intro should almost be entirely removed - the paper can stand on its own without a broad review on conspiracy theories - just focus on the interventions, that's it. Don't cite so much stuff either, just cite the Douglas et al 2019 lit review, for example.

Authors: Thank you for the suggestion – as mentioned above, we have streamlined the introduction to focus more on the interventions.

7. pg.6. Not sure that that conspiracy theory beliefs have "very real public health threats." I think the jury is still out. See:

J. Uscinski, A. M. Enders, C. Klofstad, J. Stoler, Cause and Effect: On the Antecedents and Consequences of Conspiracy Theory Beliefs. Current Opinion in Psychology https://doi.org/10.1016/j.copsyc.2022.101364, 101364 (2022).

Authors: Thank you for the comment – we have removed absolutist statement regarding the public health threats related to conspiracy beliefs and have instead highlighted that there exists an association between these two phenomenon. We also highlight that, as you noted, causal claims cannot be drawn at the moment.

8. Did the search miss "conspiratorial" because it has a t instead of a c at the end? How about misperceptions and rumors?

Authors: Our stance was that misperceptions and rumours do not fit into our definition of conspiracy beliefs. Conspiracy theories are stories that are actively created by someone/ some group to explain ambiguous events. On the contrary, misperceptions and rumours often are the result of people getting things wrong or mishearing them on the grapevine. We believe they have common traits, however, conspiracy theories are more about imposing ideological narratives to explain events.

9. There needs to be a better demarcation between general conspiracy mentality measures and specific measures of conspiracy theory beliefs. These are different things, and the implications of decreasing one versus the other are very different.

Authors: Thank you for the suggestion – we have described the differences between general and specific measures as well as referring to surrounding research that indicates that different requirements are needed to change scores on one versus the other.

10. Figure 2 is hard to read. The dots are too far from the labels.

Authors: Thank you for highlighting this –we have removed the Intervention Type label to bring the dots closer the relevant labels.

11. page 22: Conspiracy theories are not "becoming" widespread, and the literature cited, Oliver and Wood and van der linden show no such thing. Also those cites are both really old for making claims about recent over time change.

Authors: Thank you for highlighting this – We have removed this line from page 22, and we have removed any similar phrases.

12. Your paper is about conspiracy theories, not conspiracies. Be sure to use the term conspiracy theories, for example, on bottom of page 23 but elsewhere too. Conspiracies are real.

Authors: Thank you for highlighting this – this was primarily an error as we chose to focus on “conspiracy beliefs”. As such, we have removed any instances of the word “conspiracy/ies” where not appropriate.

13. Page 27. What is a "misinformation effect of conspiracy theories"? This is clunky language.

Authors: Thank you for highlighting this – we agree in retrospect that this does not read well and have instead indicated that this is “misinformation spread by conspiracy beliefs”

14. The paper, in general, needs to focus on the meta analysis, and the findings of that. Trim the front substantially and stay laser focused on the basic findings. This will highlight the paper's value. I think the abstract could highlight the main findings more: what works and what doesn't? In other words, only focus on interventions and nothing else.

Authors: Thank you for the suggestion – as mentioned above, we have streamlined the introduction to focus more on the interventions.

Thank you for your time in reviewing our work. We hope you find the paper much improved. 

 

Reviewer #4: 

This paper is a systematic review of the efficacy of interventions in countering belief in conspiracy theories. It is a very meaningful study as the field is growing fast and a review is warranted. However, the paper could benefit from some substantial changes.

Major comments:

- Because the search was conducted in June 2021, the results feel a little outdated, as with COVID-19 there is a burgeoning number of studies that look at the issue in an empirical setting, which are not included. It may be too much to ask the authors to conduct another updated search, considering the amount of anticipated work associated with it. That said, if the authors intend to keep the included papers as they are, maybe a more in-depth discussion is necessary regarding the implications for future research given all the new emerging studies.

Authors: Thank you for highlighting this – we have discussed the findings of this review in the context of emerging research in this field per your recommendation

- Relatedly, the included studies are not a lot, even though each paper investigates several interventions. The authors mentioned that many of the studies use misinformation, fake news, and conspiracy theories interchangeably. Then why not include all of them in the keyword search to expand the sample? Or at least, justify why only conspirac* is used in the search strategy.

Authors: We perhaps should have been more explicit in this section. Our initial statement sought to indicate that previous authors used fake news, misinformation, and conspiracy theories interchangeably. Many of the conspiracy interventions were developed on the basis that they were previously successful in reducing misinformation or fake news. As such, the background sections of these papers show how priming was effective by citing how it reduces fake news. However, the papers cited in the introduction themselves are focusing on conspiracy beliefs and not fake news or misinformation. We take the stance that while conspiracy beliefs are similar to fake news and misinformation, they are different things. We now specify that the conspiracy interventions listed from pages 4-6 were designed on the basis of previous findings that show their success in combatting misinformation and fake news.

- In the background section, besides explaining the different types of interventions, it's also helpful to include a discussion on the different cognitive processes of conspiracy beliefs - proportionality bias, intentionality bias, pattern perception, jumping to conclusions, confirmation bias, and the conjunction fallacy. And if possible, relate the interventions to these categories.

Authors: Thank you for the suggestion, this is an interesting idea. We have included examples of a number of cognitive biases associated with conspiracy belief, and have provided examples of how similar interventions have reduced these biases

However, the underlying processes of these interventions are still somewhat of a black box, and it is difficult to draw any conclusions regarding how these interventions relate to these cognitive biases. Based on your recommendation, we have included a section from pages 5-6 in the introduction that describe how several of the cognitive biases you mentioned have been linked to conspiracy belief. We also note how similar interventions have been successful in reducing these biases, such as the conjunction fallacy.

As a side note – for this reason, we are currently conducting a study to create a conspiracy measure that assesses how individuals critically evaluate conspiracy theories. The measure aims to identify common logical biases and fallacies associated with conspiracy beliefs related to core critical thinking domains. We aim to explore the points you mentioned above more in this coming paper.

- Is there any systematic difference between the more general and more specific conspiracy beliefs?

Authors: An interesting suggestion. We ran a independent sample t-test and found there were no significant differences between general conspiracy measures (M = 0.194, SD = 0.21) and specific conspiracy measures (M = 0.35, SD = 0.51); t(19) = -1.179, p = .25. However, as noted by Reviewer 3, previous research has indicated that specific and generic measures capture fundamentally different traits and have different implications for reducing one versus the other. As such, we have included a brief description of the differences between these measures.

- As discussed in the limitations section, the paper lacks external validity. What kind of potential interventions in real life can we devise, for example, on social media platforms?

Authors: While we would be eager to suggest a number of possibilities for real life interventions, we believe it may be too soon to draw any conclusions or make suggestion to social media platforms. For now, the jury is out regarding how we can realistically implement these interventions in real life settings. We have included in page 26 suggestions that further research should be conducted to see the viability of condensing some of the long-form interventions, and extending the effects of short-form interventions.

Minor comments:

- The PRISMA flowchart can be more detailed - breaking down the number of papers excluded according to the reason/criteria

Authors: Thank you for highlighting this – we have updated the PRISMA flow chart to meet these requirements

- Period missing at the end of page 30.

Authors: Thank you for spotting this error – we have amended the manuscript

Authors: Once again, we would like to thank you for taking the time to carefully consider our manuscript and to helpfully review our work. We hope you find that the revised manuscript is in better condition and has adequately addressed the concerns you have raised.

 

References

Bago, B., Rand, D. G., & Pennycook, G. (2022). Does deliberation decrease belief in conspiracies? Journal of Experimental Social Psychology, 103, 104395. https://doi.org/10.1016/j.jesp.2022.104395

Bierwiaczonek, K., Gundersen, A. B., & Kunst, J. R. (2022). The role of conspiracy beliefs for COVID-19 health responses: A meta-analysis. Current Opinion in Psychology, 46, 101346. https://doi.org/10.1016/j.copsyc.2022.101346

Forgas, J. P., & Baumeister, R. F. (Eds.). (2019). The Social Psychology of Gullibility: Fake News, Conspiracy Theories, and Irrational Beliefs (1st ed.). Routledge. https://doi.org/10.4324/9780429203787

Imhoff, R., Bertlich, T., & Frenken, M. (2022). Tearing apart the “evil” twins: A general conspiracy mentality is not the same as specific conspiracy beliefs. Current Opinion in Psychology, 46, 101349. https://doi.org/10.1016/j.copsyc.2022.101349

Knobel, P., Zhao, X., & White, K. M. (2022). Do conspiracy theory and mistrust undermine people’s intention to receive the COVID‐19 vaccine in Austria? Journal of Community Psychology, 50(3), 1269–1281. https://doi.org/10.1002/jcop.22714

Mulukom, V. van, Pummerer, L., Alper, S., Bai, H., Čavojová, V., Farias, J., Kay, C. S., Lazarević, L. B., Lobato, E. J. C., Marinthe, G., Banai, I. P., Šrol, J., & Žeželj, I. (2020). Antecedents and consequences of COVID-19 conspiracy beliefs: A systematic review. https://doi.org/10.31234/osf.io/u8yah

Pytlik, N., Soll, D., & Mehl, S. (2020). Thinking Preferences and Conspiracy Belief: Intuitive Thinking and the Jumping to Conclusions-Bias as a Basis for the Belief in Conspiracy Theories. Frontiers in Psychiatry, 11. Scopus. https://doi.org/10.3389/fpsyt.2020.568942

Sutton, R. M., & Douglas, K. M. (2020). Conspiracy theories and the conspiracy mindset: Implications for political ideology. Current Opinion in Behavioral Sciences, 34, 118–122. Scopus. https://doi.org/10.1016/j.cobeha.2020.02.015

---

## [Decision Letter · Decision Letter 1]

11 Jan 2023

The efficacy of interventions in reducing belief in conspiracy theories: a systematic review

PONE-D-22-25039R1

Dear Dr. O'Mahony,

We’re pleased to inform you that your manuscript has been judged scientifically suitable for publication and will be formally accepted for publication once it meets all outstanding technical requirements.

Kind regards,

Pierluigi Vellucci

Academic Editor

PLOS ONE

Additional Editor Comments (optional):

Reviewers' comments:

Reviewer's Responses to Questions

**Comments to the Author**

1. If the authors have adequately addressed your comments raised in a previous round of review and you feel that this manuscript is now acceptable for publication, you may indicate that here to bypass the “Comments to the Author” section, enter your conflict of interest statement in the “Confidential to Editor” section, and submit your "Accept" recommendation.

Reviewer #1: All comments have been addressed

Reviewer #2: All comments have been addressed

2. Is the manuscript technically sound, and do the data support the conclusions?

Reviewer #1: Yes

Reviewer #2: (No Response)

3. Has the statistical analysis been performed appropriately and rigorously? 

Reviewer #1: Yes

Reviewer #2: (No Response)

4. Have the authors made all data underlying the findings in their manuscript fully available?

Reviewer #1: Yes

Reviewer #2: (No Response)

5. Is the manuscript presented in an intelligible fashion and written in standard English?

Reviewer #1: Yes

Reviewer #2: (No Response)

6. Review Comments to the Author

Reviewer #1: (No Response)

Reviewer #2: I am happy with the changes the authors introduced in the revised version of the paper and I think it is now suitable for publication.

7. PLOS authors have the option to publish the peer review history of their article (what does this mean?). If published, this will include your full peer review and any attached files.

Reviewer #1: No

Reviewer #2: No

---

## [Editor Report · Acceptance letter]

14 Mar 2023

PONE-D-22-25039R1 

The efficacy of interventions in reducing belief in conspiracy theories: a systematic review 

Dear Dr. O'Mahony:

I'm pleased to inform you that your manuscript has been deemed suitable for publication in PLOS ONE. Congratulations! Your manuscript is now with our production department. 

Kind regards, 

on behalf of

Dr. Pierluigi Vellucci 

Academic Editor

PLOS ONE